

# The role of rock fractures on tree water use of water stored in bedrock: Mixing and residence times

Xiuqiang Liu[1], Xi Chen[1,2], Zhicai Zhang[3], Weihan Liu[1], Tao Peng[4,5], Jeffrey J. McDonnell[6,7,8]

[1]School of Earth System Science, Institute of Surface-Earth System Science, Tianjin University, Tianjin 300072, China

[2]Tianjin Key Laboratory of Earth Critical Zone Science and Sustainable Development in Bohai Rim, Tianjin University, Tianjin 300072, China

[3]College of Hydrology and Water Resources, Hohai University, Nanjing 210098, China

[4]State Key Laboratory of Environmental Geochemistry, Institute of Geochemistry, Chinese Academy of Sciences, Guiyang 550081, China

[5]Puding Karst Ecosystem Research Station, Chinese Academy of Sciences, Puding 562100, China

[6]Global Institute for Water Security, School of Environment and Sustainability, University of Saskatchewan, Saskatoon, Saskatchewan, Canada

[7]North China University of Water Resources and Electric Power, Zhengzhou, China

[8]School of Geography, Earth and Environmental Sciences, University of Birmingham; Birmingham, UK.

*Corresponding to*: Xi Chen (xi_chen@tju.edu.cn)

**Abstract.** The processes of tree water uptake in karst environments are poorly understood. One of the main challenges to improved understanding is the complex interaction between soil water and bedrock water, especially in systems characterized by rock fractures. While some studies have highlighted the potential importance of fractured bedrock as a water source for plants, few have quantitatively assessed the sources, residence times, and seasonal dynamics of tree water uptake from both soil and rock fracture compartments. Here, we combine stable isotope tracing, a Bayesian mixing model (MixSIAR), and hydrometric monitoring to quantify the contributions and mean residence times (MRT) of soil and rock water accessed by trees across seasons. We use a four-compartment sampling framework that distinguishes between soil water (mobile and bulk) and rock water (fracture and infilled fracture). Our results reveal clear seasonal shifts in plant water sourcing: during the peak rainy season, mobile soil water (mean MRT = 88 days) dominates uptake (mean contribution 41%), whereas in late growing season, trees increasingly rely on bulk soil water (mean MRT = 95 days, mean contribution 55%). Strikingly, in early spring, trees in fracture-rich areas exhibit the highest reliance on rock water (mean MRT = 113 days, mean contribution 69%). During the subsequent early growing season, large trees derive up to 85% of their water from rock, particularly from soil-filled fractures with apertures >10 mm, which act as transitional reservoirs capable of retaining precipitation for extended periods (MRT = 84-303 days). Trees preferentially access short-MRT sources under wet conditions and shift to longer-MRT pools during dry periods, reinforcing the concept of ecohydrological separation between tightly bound and dynamically recharged water pools. However, this separation is attenuated during periods of high precipitation due to increased hydraulic connectivity and water mixing. This work advances our understanding of vegetation resilience in structurally complex and hydrologically dynamic karst landscapes with important insights for sustainable water resource management under changing climatic conditions.

## 1 Introduction

Water transport and retention are critical for understanding the hydrodynamic behavior of the soil-plant-atmosphere



continuum, in which the transfer of water through the plant and soil is a central process (Philip, 1966; Deng et al., 2017). Climate and critical zone regimes influence how plants access and utilize water resources by modulating precipitation inputs, root zone water transit, and residence times (Luo et al., 2023; Fan and Miguez-Macho, 2024; Liu et al., 2024c). The time it takes for water to be transported within trees can affect how forests store and release water, thereby altering the timing and
magnitude of water fluxes such as transpiration and runoff at the ecosystem scale (Bond et al., 2008).

Stable isotopes of hydrogen and oxygen ($^2$H and $^{18}$O) have been powerful tracers for quantifying water transport (Putman and Bowen, 2019). When combined with isotope-based models, such as linear mixing models (IsoSource, SLM), Bayesian mixing models (SIAR, MixSIR, and MixSIAR), and continuous distribution models (e.g., CrisPy), the relative contributions of each water source to plant water use can be quantified (Putman and Bowen, 2019; Wang et al., 2019a; Fu et al., 2024). These
tools have been widely applied to identify when and where soil water is accessed by plants (Midwood et al., 1998; Tang and Feng, 2001; Darrouzet-Nardi et al., 2006; Mccole and Stern, 2007; Brooks et al., 2010; Wang et al., 2010). Recent advances have introduced the "ecohydrological separation hypothesis", which posits that water contributing to plant transpiration is often distinct from the water that generates runoff or groundwater recharge (Brooks et al., 2010; Mcdonnell, 2014). This framework suggests the coexistence of at least two functionally distinct water reservoirs within the soil: a mobile, hydrologically connected
fraction and a more strongly bound, plant-available fraction (Sprenger and Allen, 2020; Finkenbiner et al., 2022).

However, beginning with work of Salve et al. (2012) and Oshun et al. (2016) increasing attention has been given to a third, hidden water reservoir: rock moisture, which refers to water stored in unsaturated weathered bedrock (Rempe and Dietrich, 2018; Zhang and Zhang, 2021). This water often exists at matric potentials above the turgor loss point of roots (> -1.5 MPa) (Zwieniecki and Newton, 1996; Schoeman et al., 1997; Hubbert et al., 2011; Korboulewsky et al., 2020; Nardini et
al., 2021). Under prolonged drought conditions in certain environments, plants can shift their water use strategy from shallow soil water to this deeper rock water (Barbeta et al., 2015; Hahm et al., 2022; Jiménez-Rodríguez et al., 2022; Ning et al., 2023; Nardini et al., 2024). But while considerable progress has been made in understanding the role of rock moisture and its geographical extent (Mccormick et al., 2021), the mechanisms of rock moisture extraction are still poorly understood. Since much of the weathered rock zone is fractured, few if any studies have quantified the role of rock fracture water in plant water
sourcing from rock moisture.

We know that plant roots have the ability to access rock water in fractures (Schwinning, 2020). Studies suggest that the minimum fracture aperture required for root water uptake is approximately 0.1 mm (Zwieniecki and Newton, 1995; Schwinning, 2010). Once this threshold is exceeded, capillary forces diminish, while both hydraulic conductivity and volumetric flow increase significantly with fracture aperture (Zimmerman and Bodvarsson, 1996; Wang et al., 2015; Wang et
al., 2022; Liu et al., 2024b). These hydraulically active microfractures can establish water connectivity across fracture networks (Wolfsberg, 1997), forming dynamic reservoirs (Vrettas and Fung, 2017; Jiménez-Rodríguez et al., 2022; Ning et al., 2023), that enhance the likelihood of plant access to rock moisture (Schwinning, 2010, 2020). Moreover, these fractures can act as sheltered conduits for root extension, buffering roots against environmental extremes such as thermal fluctuations and wind exposure, while their enclosed structure reduces evaporative losses, allowing water to be retained for longer periods (Pawlik



et al., 2016; Zhang et al., 2016; Preisler et al., 2019; Hahm et al., 2020; Luo et al., 2024a). This protective effect allows roots to penetrate deep into the substrate and access water and nutrients that are inaccessible in shallow soils (Zhang et al., 2019a). In many cases, larger fractures may become partially or fully filled with soil, further enhancing their capacity to retain water and provide nutrients to vegetation (Estrada-Medina et al., 2013; Yang et al., 2016; Peng et al., 2019; Liu et al., 2024b).

However, the mechanistic controls underlying plant water uptake from different fracture types remain poorly understood.
Plants growing on different rock substrates may access varying amounts of available water, depending on the physical properties of the underlying fractures (Zwieniecki and Newton, 1996; Querejeta et al., 2006; Nardini et al., 2021; Nardini et al., 2024). No study that we are aware of has yet systematically examined how fracture aperture, depth, and degree of soil infill influence both the residence time of stored water and the patterns of plant water uptake. These structural attributes may determine whether a fracture functions as a fast conduit, a temporary buffer, or a long-term reservoir for plant-available water.

Here we conduct a systematic and mechanistic examination of the role of rock fractures on tree water use of water stored in bedrock. We include within-fracture measurements and sampling and focus on a karst landscape. In such karst environments epikarst has been shown to play a pivotal role as a hydrological interface between surface and subsurface systems and potentially regulates water storage and transmission, as well as plant growth (Wang et al., 2022). Epikarst is known to range from 0.5 to 30 meters (Williams, 2008), and its highly developed pore and fracture networks not only facilitate the
transformation of surface water into groundwater but also provide growth pathways for plant roots to access deep rock fractures. We focus on a site in southwest China's karst region that is characterized by a subtropical monsoon climate with distinct seasonality. Here summer precipitation is abundant, the shallow soil layer, often less than 50 cm thick, but the high permeability of the fracture network result in rapid water percolation and shortened residence times of moisture within fractures (Zhang et al., 2019b; Jiang et al., 2020), ultimately increasing the frequency of drought events (Wang et al., 2019b; Xu et al., 2023).
Nevertheless, the structure of the soil-rock system can modulate local water storage and availability. In areas where the epikarst has a stronger water-holding capacity, the combined soil-fracture system can retain both water and nutrients, offering critical support to karst ecosystems and maintaining their stability and sustainability (Wang et al., 2024b).

We hypothesize that subsurface partitioning may extend beyond the soil matrix to include fractured bedrock, where water is stored in forms ranging from fast-draining open fissures to long-residence, soil-filled fractures. Although these subsurface
compartments are hydraulically connected, they may differ substantially in renewal rates, retention times, and plant accessibility, effectively forming a structurally and functionally analogous extension of ecohydrological separation. This study aims to provide a scientific foundation for sustainable water resource management and vegetation restoration in karst environments. Specifically, this study will address three fundamental research questions:

1. How do rock fractures (aperture size, filling conditions, hydraulic properties) affect water uptake patterns of plant roots?
2. What factors influence the residence time and transit times in soil, rock and plant water system in karst?
3. To what extent does ecohydrological separation occur and seasonal rainfall play in shaping this separation?

To address these questions, we use a sampling approach where soil water is subdivided into mobile water (sampled via lysimeters) and bulk water (sampled via cryogenic extraction), while rock water comprises rock fracture water and infilled



rock fracture water. This enabled us to distinguish among four types of water sources: mobile soil water, defined as water
residing in connected soil pores that responds rapidly to rainfall events; bulk soil water, defined as more tightly bound water
in micropores, characterized by slower renewal and longer retention; rock fracture water: defined as water stored in clean,
unfilled fractures within weathered bedrock, often transient and dynamic and infilled rock fracture water, defined as water held
in fractures partially or fully filled with soil, offering higher retention and potential nutrient supply.

## 2 Material and methods

### 2.1 The study area and sampling location

Our experimental site is located at the Puding Karst Ecosystem Research Station in Guizhou, southwest China (105°42'-
105°43'E, 26°14'-26°15'N) (Fig. 1). The climate is dominated by subtropical humid monsoon. The mean annual temperature
is 15.1°C, and the mean annual precipitation is 1378 mm. Over 80% of annual precipitation occurs from May to October as
summer rainfall (Liu et al., 2022). The carbonate rocks consist mainly of limestone. Soils developed from black and yellow
limestone are distributed irregularly across the study site (Li et al., 2023) with soil-filled 'grykes' (i.e. cracks that form
vertically in the limestone via chemical weathering, Parry (1960)) widely distributed (Fig. 1). The thickness of the root-zone
soil ranges from 15 cm to 200 cm (mean: 66 cm), the bare rock coverage varies from 0.42 to 0.94, and the apertures of rock
fractures range from 0.1 mm to 38 mm (mean: 2.4 mm), with over 70% of fractures having apertures narrower than 1 mm.
Fractures with apertures greater than 10 mm are often completely or partially filled with soil (Liu et al., 2024b; Liu et al.,
2025). The vegetation at the site consists of secondary broadleaf forests that have regenerated after environmental disruption
caused by human activities, such as deforestation for cultivation.

We selected five typical sites (A-E in Fig. 1) to survey the soil-rock structure and tree stand structure (Liu et al., 2025). The
five sites included 18 individual trees of 5 different species, including deciduous broadleaf trees (*Broussonetia papyrifera*
(L.) L'Hér. ex Vent. ($bp_{1-5}$), *Koelreuteria paniculata* Laxm. ($kp_{1-2}$), *Toona sinensis* (A.Juss.) M.Roem. ($ts_{1-6}$), and *Yulania*
*denudata* (Desr.) D.L.Fu. ($yd_{1-6}$)), and one evergreen broadleaf tree (*Cinnamomum camphora* (L.) J.Presl. ($cc_{1-3}$)).



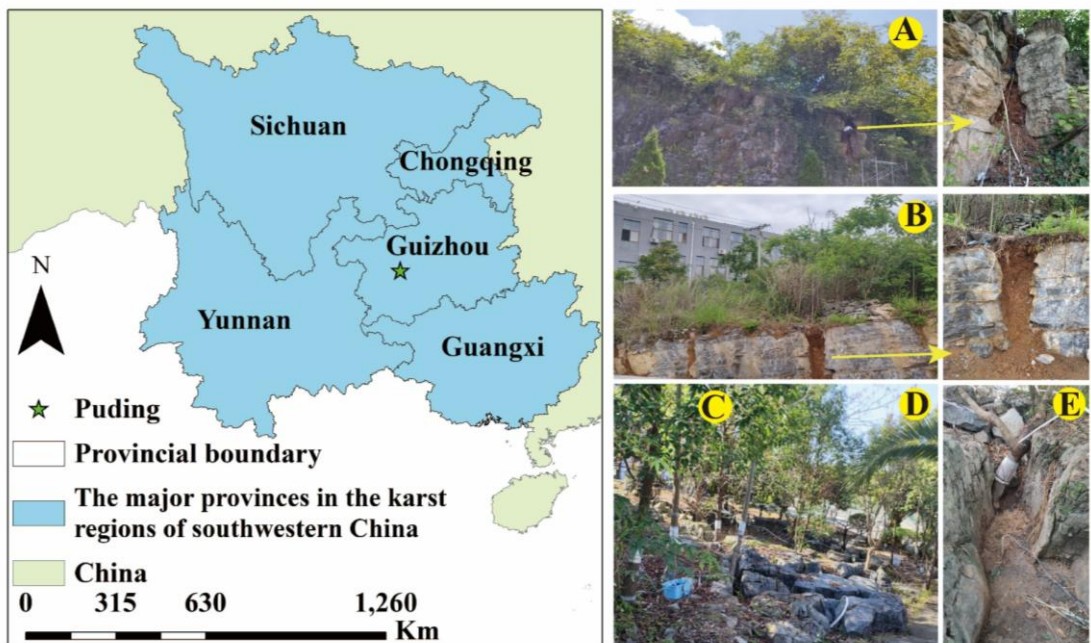

**Figure 1. The study area and five sampling sites.**

## 2.2 Field measurements

### 2.2.1 Collecting samples for isotope analysis

The method for collecting mobile soil water and bulk soil water samples at different depths involved two preparatory steps: (1) Using a soil auger, we manually drilled soil boreholes to depths of 0.1 to 2 m; (2) At each target depth, soil was removed, and ceramic tensiometer tips connected to suction lysimeters were inserted. The boreholes were then backfilled with the original soil to ensure proper contact and minimize disturbance.

To determine the suction range corresponding to mobile soil water, this study constructed soil water retention curves

based on previously established soil hydraulic parameters (Liu et al., 2024b). According to the soil moisture distribution characteristics, the mobile water content range and its corresponding maximum suction for different soil types were identified. The results indicate that the maximum suction range of mobile water at different sample sites is as follows: Site A: 70-90 kPa, Site B: 65-90 kPa, Sites C&D: 40-45 kPa, and Site E: 55-80 kPa. Therefore, during the actual extraction of mobile soil water, it is necessary to set the applied vacuum values appropriately based on the hydraulic characteristics of different soil layers,

ensuring that they are below the maximum suction range of mobile water. This practice ensures that the collected water mainly originates from mobile soil water, avoiding excessive extraction of bound water, thus enhancing the representativeness and accuracy of the sampling. This method is also consistent with the findings of Sprenger et al. (2018), who noted that mobile soil water collected by vacuum samplers typically corresponds to a maximum suction of about 60 kPa.

Bulk soil water was extracted from the collected soil samples through laboratory extraction.

For the collection of rock fracture water, 26 representative fracture locations were selected based on prior field



investigations (Liu et al., 2024b). At each location, boreholes were drilled and rock cores were extracted using a portable core drilling tool. We inserted the ceramic tensiometer tips with PVA sponges into the boreholes so that the ceramic tips were located at the fractures where water samples needed to be collected. We used sponges to isolate other fracture layers to prevent mixing of fracture water from different depths. The ceramic tips in the soil and fractures were connected to sampling bottles, and a
vacuum pump was used to create a negative pressure in the sampling bottles to collect rock water.

For collecting tree branch samples we removed the bark, keeping only the xylem, and sealed them promptly. All soil and xylem samples were stored frozen, while soil water, fracture water, and rainwater samples were refrigerated for storage.

From September 2022 to March 2025, samples were collected at an average frequency of 1-3 times per month.

### 2.2.2 Moisture and sap flow monitoring

Meteorological variables were observed at the standard meteorological station of the Puding Karst Ecosystem Research Station. Automatically recorded variables included precipitation ($P$), air temperature at 2 m height ($T_a$), and soil moisture content at depths of 0-200 cm at each sampling site, monitored using capacitance-based frequency domain reflectometry (FDR) sensors (5TM, METER Group, Pullman, WA, USA) at 30 minute intervals. Additional sensors at various depths were also installed at the soil-rock interface profiles at sites A and B. Detailed procedures for monitoring rock moisture content and sap
flow in trees are described in (Liu et al., 2025).

### 2.3 Laboratory analyses

Sample processing and testing were divided into two periods. Samples collected from September 2022 to March 2025 were processed and analyzed at the Ecohydrology and Water Resources Research Center and the Experimental Test Analysis Science and Technology Center of the School of Earth System Science at Tianjin University.

To collect xylem water from vegetation, the collected arboreal branches were processed using the LI-2100 fully automatic vacuum condensation extraction system (Beijing Ligajoint Scientific Co., Ltd., China) to extract liquid water (cryogenic vacuum distillation technique, CVD). Prior to extraction, the samples were weighed with a microbalance (accuracy of 0.1 mg), and the heating temperature for extraction was set to 150°C with a duration of three hours. After extraction, the samples were weighed again to calculate the volume of water extracted, and then dried at 105°C for 24 hours. The dry samples were weighed
again to calculate the total water loss, and the water extraction rate was calculated based on the weight loss (the proportion of water extracted to the total water loss after drying). The water extraction rate for all samples was generally above 99%. Samples with less than 99% water extraction rate were re-extracted until the standard was met.

Water isotopes δD and δ$^{18}$O in rainfall, mobile soil water, and rock water samples were measured using the Picarro L2140-i water isotope analyzer. The measurement errors for δD were 0.1‰, and for δ$^{18}$O were 0.015‰. As tree branch water contains
volatile organic compounds (VOCs), all plant extracted waters were analyzed using the MAT 253 Plus gas stable isotope ratio mass spectrometer to determine δD and δ$^{18}$O values. The errors for δD were 2‰, and for δ$^{18}$O were 0.2‰.

Samples from April 2024 to March 2025 were processed and analyzed at the University of Saskatchewan in Canada.



Water was extracted from plant xylem and soil samples using the CVD method developed by Koeniger et al. (2011). After freezing the sample vials with liquid nitrogen, they were connected via a capillary tube to a collection vial, vacuumed to below 0.8 mbar, and heated at a predetermined temperature for a specified duration. In this closed system, vaporized water from the sample vial was condensed into the collection vial. The extracted water was then transferred to 2 mL screw-cap glass vials. Extracted samples were subsequently oven-dried at 105 °C for 24 hours to evaluate extraction efficiency. Plant samples were extracted at 200 °C for 24 minutes, while soil samples were extracted at the same temperature for 30 minutes (Wang et al., 2024a). Higher extraction temperatures and efficient water recovery help to minimize isotope ratio errors in soil and xylem water samples (Younger et al., 2024).

Rainfall, mobile soil water, and rock water samples were analyzed using an Off-Axis Integrated Cavity Output Spectroscopy (OA-ICOS) system from Los Gatos Research, with post-processing performed via the LIMS for Lasers 2015 software. Due to the presence of organic compounds in plant and soil-extracted water samples, these were analyzed using isotope ratio mass spectrometry (IRMS). For hydrogen isotope analysis, the elemental analyzer-IRMS (EA-IRMS) method was employed, where water samples reacted with elemental chromium at high temperatures to produce hydrogen gas, following the method of Morrison et al. (2001). The resulting hydrogen gas was separated using a gas chromatographic column and then analyzed by IRMS. For oxygen isotope analysis, the $CO_2$-$H_2O$ equilibration method described by Epstein and Mayeda (1953) was used Epstein and Mayeda (1953), wherein water equilibrates with $CO_2$ at a controlled temperature and the isotopic composition of the equilibrated $CO_2$ is then measured by IRMS.

The analytical uncertainties ($2\sigma$) for $\delta D$, $\delta^{18}O$, and $\delta^{17}O$ were ± 2‰, ± 0.8‰, and ± 0.5‰, respectively. Laboratory reproducibility was ± 1.0‰ for $\delta D$ and ±0.2‰ for both $\delta^{18}O$ and $\delta^{17}O$. All isotope values were reported in per mil (‰) relative to the Vienna Standard Mean Ocean Water-Standard Light Antarctic Precipitation (VSMOW-SLAP) scale. To ensure data quality, selected samples were analyzed in both laboratories, and the results were highly consistent, confirming the reliability of the measurements.

**2.4 Data analysis**

**2.4.1 Estimation of mean transit time and mean residence time**

The mean residence time (MRT) represents the average time during which water remains in a particular compartment before moving out. For example, 50 cm of soil may retain infiltrated rainfall for a certain period before it contributes to deeper flow or evaporation. We estimated MRT using the amplitude damping approach (Małoszewski and Zuber, 1982; Stewart and Mcdonnell, 1991; Mcguire et al., 2002; Reddy et al., 2006). Amplitude damping reflects the attenuation of isotopic signal fluctuations due to processes such as dilution and mixing during water transport.

The isotope sine curve analysis is used to estimate MRT by fitting a sinusoidal function to the isotope data of each compartment. We used the sine function fitting:

$$\delta^{18}O \text{ or } \delta D = a_1 \cos(2\pi t) + a_2 \sin(2\pi t) + offset \tag{1}$$

Where , $a_1$ and $a_2$ are the amplitude parameters, and *offset* represents the baseline value.

We applied a sine function transformation to the isotopic time series ($\delta^{18}O$ or $\delta D$) as follows:





$$\delta^{18}O \ or \ \delta D = A \cdot \sin(2\pi t - \phi) + offset \tag{2}$$

Amplitude ($A$) and phase ($\phi$) were calculated as:

$$A = \sqrt{a_1^2 + a_2^2} \tag{3}$$

$$\phi = -atan2(a_1, a_2) \tag{4}$$

The MRT (in days) was calculated by:

$$MRT = c^{-1}\sqrt{f^{-2} - 1} \tag{5}$$

$$f = \frac{A_n}{A_m} \tag{6}$$

Where $A_n$ and $A_m$ are the amplitudes of the output and input isotope signals, respectively. The damping factor ($f$) was used to describe how the amplitude of the isotope signal is reduced as it moves through different compartments.

### 2.4.2 MixSIAR model

To analyze the partitioning of water sources among plants in our karst ecosystem, we employed the MixSIAR Bayesian Mixing Model (BMM). By integrating isotopic data, we gained a comprehensive understanding of the contributions of various water sources to plant water uptake. MixSIAR is a stable isotope analysis package in R that estimates water source contributions through a Bayesian approach, using Markov Chain Monte Carlo (MCMC) methods to generate posterior distributions of source proportions. The MCMC algorithm runs on multiple chains to ensure convergence and reliable posterior estimates (Moore and Semmens, 2008; Parnell et al., 2013; Stock and Semmens, 2016). One of the main advantages of MixSIAR over traditional mass balance methods is its reduced sensitivity to isotopic fractionation effects (Evaristo et al., 2017), and it incorporates uncertainties from the sources by introducing prior information, multiple continuous covariates, and error structures to improve prediction accuracy (Gai et al., 2023; Liu et al., 2024a). This robustness is particularly well-suited to karst ecosystems, where the isotopic composition of water sources can vary significantly due to complex interactions among soil, rock fractures, and vegetation (Gai et al., 2023).

### 2.4.3 Calculation Method of Evaporation Index

The evaporation degree of different water bodies relative to local precipitation is characterized using the linear offset (Lc-excess) from the Local Meteoric Water Line (LMWL), as proposed by Landwehr and Coplen (2006):

$$Lc - excess = \delta D - a \times \delta^{18}O - b \tag{7}$$

where $a$ and $b$ are the slope and intercept of the LMWL, respectively.

## 3. Results

### 3.1 Temporal variation in isotope values and water conditions within water pools

Fig. 2 shows that the $\delta^{18}O$ and $\delta D$ values of precipitation exhibit periodic sinusoidal fluctuations with the seasons, closely related to changes in precipitation amount and temperature. During the summer and autumn months (June to September), when rainfall is more abundant, the precipitation primarily originates from maritime air masses, with noticeably more negative $\delta^{18}O$ and $\delta D$ values, indicating the influence of the amount effect. Moreover, after intense rainfall events, $\delta^{18}O$ and $\delta D$ values



decrease sharply in a short period, reflecting the significant impact of intense rainfall on the local hydrological cycle,

suggesting that such precipitation may originate from the input of atmospheric water vapor from higher altitudes. During the winter and spring months (December to April), precipitation is mainly controlled by continental air masses, with relatively higher $\delta^{18}O$ and $\delta D$ values, reflecting isotopic enrichment characteristics caused by lower rainfall amounts and stronger evaporation effects.

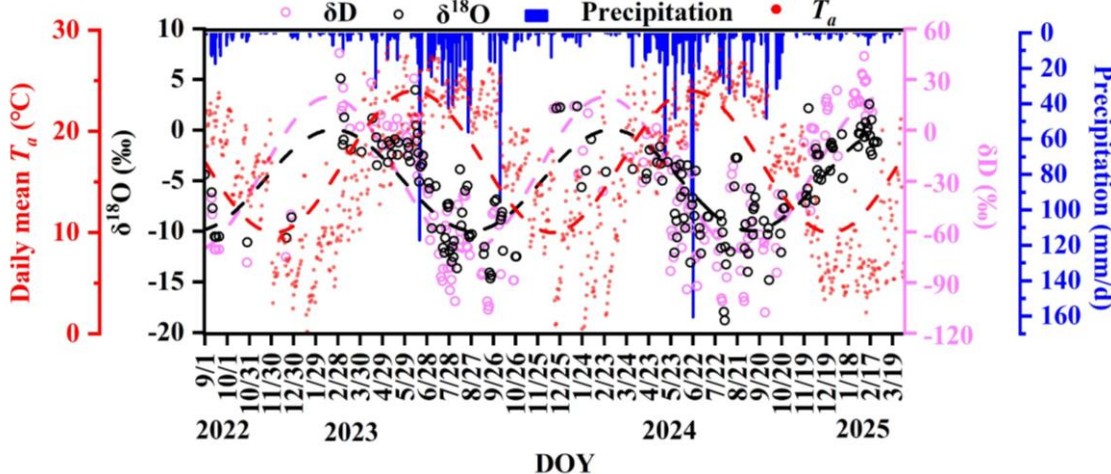


**Figure 2. Temporal variations in stable isotopes of precipitation ($\delta D$ and $\delta^{18}O$), rainfall amount, and daily average temperature.**

      Fig. 3 indicates that the average $\delta^{18}O$ value of rainfall in the study area is -6‰, and the average $\delta D$ value is -35.66‰, with a wide range of variations. It presents the LMWL for the study area, with a slope of 8 and an intercept of 9.93, which is

consistent with the Global Meteoric Water Line (GMWL), indicating that the precipitation isotopes in the study area are mainly controlled by the source of precipitation and the amount effect, without significant non-equilibrium evaporation during atmospheric transport.

      Fig. 3 indicates that there are significant differences in the $\delta^{18}O$ and $\delta D$ values of mobile soil water and bulk soil water across various sampling sites in the study area. The average $\delta^{18}O$ for mobile soil water is -6.57‰, with average $\delta D$ is -43.79‰,

showing a broad overall range ($\delta^{18}O$: -16.99 to 1.27‰; $\delta D$: -127.87 to 5.89‰). Appendix Figs. A1a and A1b depict the distinct sinusoidal fluctuations in $\delta^{18}O$ and $\delta D$ for mobile soil water, which become enriched after the reduction in precipitation in September and shift towards more negative values following increased precipitation in April, aligning with the isotopic characteristics of precipitation.

      In contrast, bulk soil water exhibits lower isotopic values, with average $\delta^{18}O$ is -8.13‰ and average $\delta D$ is between -

59.61‰. Fig. 3 shows that the overall evaporation line slope of bulk soil water (8.21) aligns with the precipitation line and is higher than that of mobile soil water (7.53). The isotopic trends of bulk soil water also show sinusoidal fluctuations, with shallow soil water (0-30 cm) affected more significantly by precipitation and evaporation, showing more pronounced



fluctuations, while deeper soil water (> 30 cm) changes more gradually. Additionally, Appendix Fig. A1c shows that shallow soil moisture content fluctuates significantly, especially at the surface 5 cm, while deeper soil moisture remains relatively

stable. Soil moisture content is highest at depths of 20-50 cm and lowest below 50 cm.

Soil-rock interface water have lower averages ($\delta^{18}O$ = -6.77‰, $\delta D$ = -46.69‰) than mobile soil water but higher than bulk soil water. Appendix Fig. A1d shows that soil-rock interface moisture content responds more rapidly and with greater fluctuation to rainfall than the soil at the same depth, indicating that the soil-rock interface is not fully continuous but has preferential flow pathways, acting as rapid water transmission channels during rainfall infiltration, thus aligning its

hydrological dynamics more closely with rock fracture water rather than the slow response patterns of surrounding soil water.

The mean $\delta^{18}O$ and $\delta D$ values of rock fracture water were -6.89‰ and -44.39‰, respectively, while those of infilled rock fracture water were -6.73‰ and -46.11‰. However, the overall evaporation line slope for rock fracture water (8.31) was greater than that of infilled rock fracture water (6.76). Appendix Figs. A2 further reveal isotopic and moisture characteristics of rock fracture water under different fracture apertures and depths. For instance, at site B, shallow fracture water (45-60 cm)

$\delta^{18}O$ and $\delta D$ values (Figs. A2a and b) and moisture content (Fig. A2c) fluctuate significantly, indicating primary influence by short-term rainfall, whereas at site C, deep fracture water (244-306 cm, narrower aperture 0.16-0.26 mm) shows smaller fluctuations in $\delta^{18}O$, $\delta D$, and moisture content, indicating slower water renewal rates. In larger fractures filled with soil, such as at sites B (10-21 mm) and C (38 mm), the isotopic values and moisture content of fracture water show smaller fluctuations compared to adjacent depths, suggesting that soil filling reduces the impact of rapid infiltration from rainfall, making the water

replenishment process more stable.

The mean $\delta^{18}O$ and $\delta D$ values of xylem water were -5.99‰ and -51.12‰, respectively. The relatively depleted $\delta D$ and the overall evaporation line slope of 6 suggest a deviation below the LMWL, indicating stronger evaporative enrichment.

Appendix Figs. A3a and A3b show that the $\delta^{18}O$ and $\delta D$ values of xylem water in trees exhibit seasonal fluctuations similar to those of the water sources, and the $\delta^{18}O$ and $\delta D$ values of various water sources at different sites show certain

similarities, directly influencing the isotopic characteristics of tree xylem water. For example, water sources at site B are overall more enriched, while those at sites A and C&D are more depleted, leading to more enriched isotopic values in xylem water at site B (average $\delta^{18}O$ -5.50‰, average $\delta D$ -49.58‰) compared to site A ($\delta^{18}O$ -6.04‰, $\delta D$ -51.57‰) and sites C&D ($\delta^{18}O$ -6.17‰, $\delta D$ -51.90‰). This difference indicates that the water use by vegetation is directly controlled by the hydrological conditions at different sites.






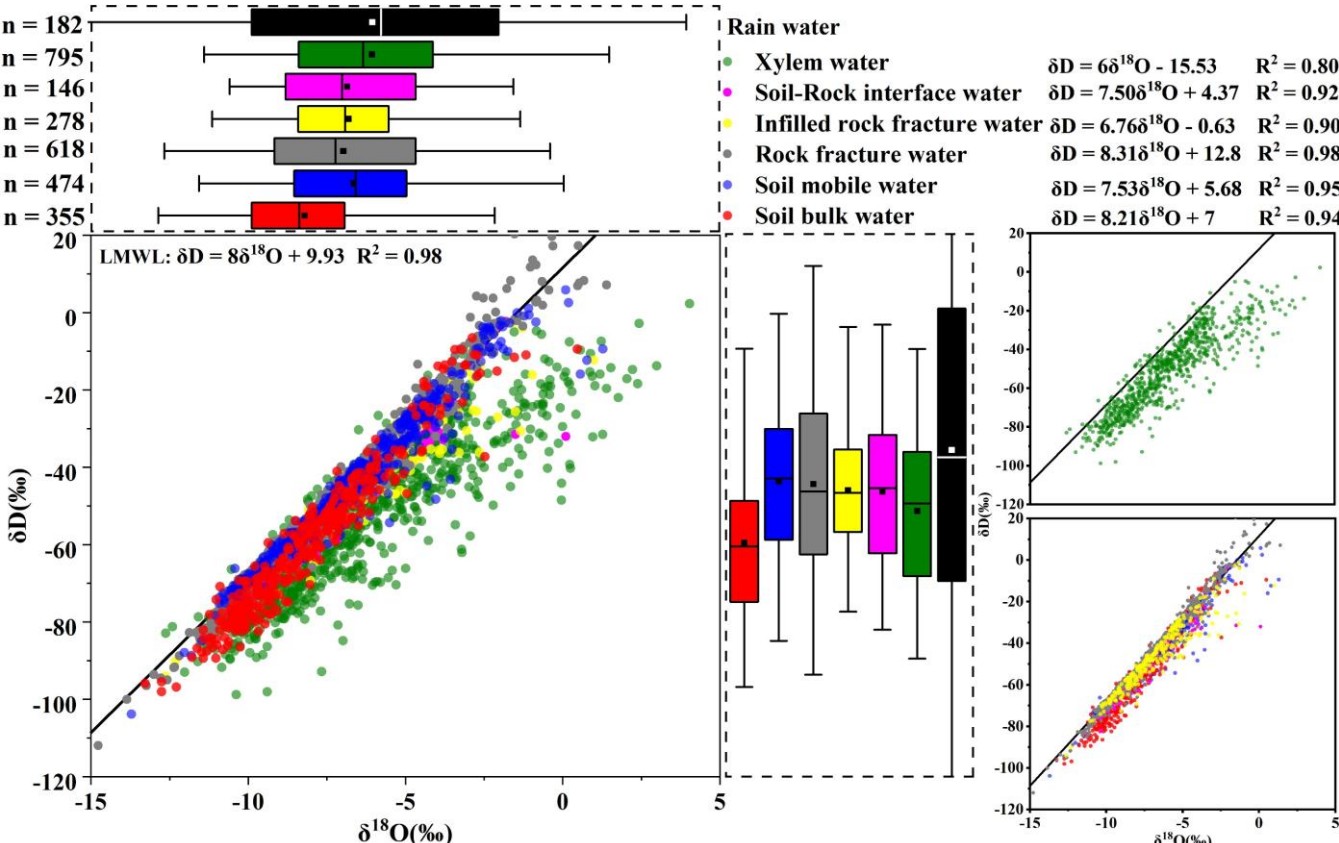

**Figure 3. δD-δ¹⁸O relationship diagram for different water bodies during various periods.**

## 3.2 Factors affecting MRT in different water pools

Appendix Table B1 lists the MRT for mobile soil water and bulk soil water at different sample sites and soil depths. The MRT of soil water at different sites are influenced by a combination of soil depth and permeability, showing distinct spatial distribution characteristics. Overall, soil water MRT ranged from 36 to 708 days, with an average of 177 days. Specifically, the MRT of bulk soil water (36-708 days, mean 196 days) was higher than that of mobile soil water (63-459 days, mean 143 days). Similarly, by referencing tree root depths from Table B4 and the corresponding soil water MRTs from Table B1, we

found that bulk soil water within the root zone (36-177 days, mean 95 days) exhibited a longer residence time than mobile soil water (63-144 days, mean 88 days).

   The MRT at site B (49-708 days, mean 365 days) was significantly higher than that at other sites (36-144 days, mean 88 days), indicating substantial differences in water retention characteristics among water types and sampling locations. With increasing soil depth, MRT of bulk soil water increased markedly. For example, at site B, MRT ranged from 49 to 177 days

(mean 123 days) at depths of 5-20 cm. When depth exceeded 50 cm, MRT increased to 244-708 days (mean 513 days), suggesting that deep soil water has significantly longer residence times and lower transport rates. In contrast, the MRT of mobile soil water gradually decreased with depth from 50 to 180 cm. This suggests that deep mobile water may be replenished





by preferential flow along soil-rock interfaces or fractures. The opposing depth-dependent MRT patterns between mobile and bulk soil water indicate their different hydrological functions. Bulk soil water serves as a stable storage with slow turnover, whereas mobile soil water, functions as a transient reservoir connected to rapid flow paths such as fractures.

Soil $K_h$ also played a crucial role in controlling MRT. As shown in Fig. 4, MRT of bulk soil water decreased exponentially with increasing $K_h$. When $K_h$ exceeded 0.75 cm·h$^{-1}$, MRT stabilized around 110 days. In deep soils (> 50 cm) at site B, $K_h$ values ranged only from 0.1 to 0.18 cm·h$^{-1}$, substantially lower than those at other sites (0.75-2.33 cm·h$^{-1}$) which severely restricted water movement and resulted in significantly elevated MRT values. This indicates that in low-permeability soils, water has a longer residence time and slower renewal rate, whereas in high-permeability soils, water is replenished more rapidly, resulting in shorter MRT.

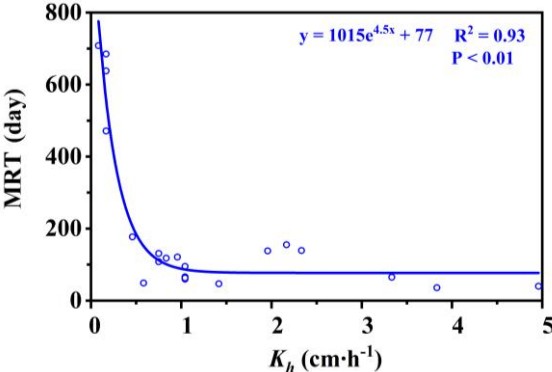

**Figure 4. Relationship between $K_h$ and MRT of bulk soil water.**

Table B2 shows that at site B, the MRT of soil-rock interface water (21-214 days, mean 118 days) were significantly lower than that of soil water at the same depth (244-708 days, mean 513 days; see Table B2). These results indicate that the soil-rock interface has higher permeability and more developed preferential flow pathways, leading to a faster renewal rate of water at the interface. Consequently, soil-rock interface water is more responsive to precipitation inputs but has lower retention capacity.

Table B3 presents the MRT of rock fracture water under different fracture apertures and burial depths. Overall, MRT ranged from 23 to 303 days (mean 117 days), showing pronounced spatial heterogeneity influenced by fracture aperture, depth, and recharge conditions. For shallow fractures (burial depth 45-80 cm, aperture 0.47-2 mm), MRT ranged from 23 to 91 days, indicating that shallow fracture water is strongly influenced by precipitation and has a high renewal rate. At site A, deep fractures (270-600 cm) exhibited MRTs of 108-182 days, while fractures at sites C&D (244-306 cm depth) showed MRTs of 118-124 days. This suggests that deep fracture water is primarily recharged by old water, resulting in slower renewal and a more damped isotopic signal. A significant relationship was observed between MRT and fracture aperture (Fig. 5). Notably, when fracture width exceeded 10 mm, MRT increased substantially with aperture size, ranging from 84 to 303 days (mean 172 days). This is primarily due to the presence of soil infill in large fractures (porosity 0.41-0.74), which enhances the water-



holding capacity and prolongs water retention time. This process has important implications for plant water use: infilled rock
fractures can provide a more stable water supply under drought conditions, thereby increasing water availability and potentially
influencing the water uptake strategies of different plant root systems (Zhang et al., 2016; Hasenmueller et al., 2017; Carrière
et al., 2019; Howarth and Bishop, 2023; Yan et al., 2023).

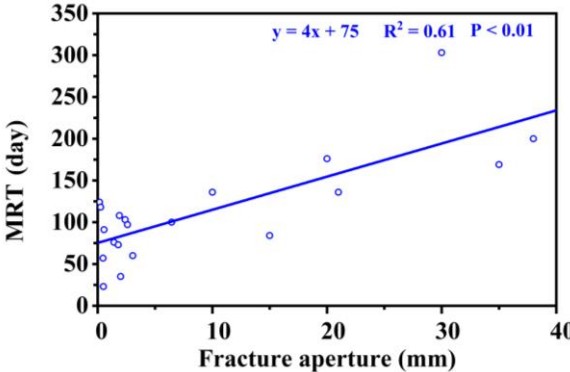

**Figure 5. Relationship between aperture and MRT of rock water.**

Table B4 details the MRT for different water sources in various trees, reflecting the time it takes for mobile soil water,
bulk soil water, and rock water to be absorbed from the rhizosphere to the tree interior, as well as the duration these water
sources stay within the trees.

There are significant differences in the MRT of different water sources within the trees at different sites, indicating spatial
heterogeneity in plant water use strategies. At site A, the differences in MRT among the three water sources are relatively
minor ($MRT_1$ ranges from 39 to 59 days, $MRT_2$ from 44 to 64 days, $MRT_3$ from 31 to 53 days), suggesting that trees at this
site have a flexible water use strategy. They can simultaneously utilize mobile soil water, bulk water, and rock fracture water
to adapt to spatial and temporal variations in water supply. In contrast, at site B ($MRT_1$ ranges from 14 to 40 days, $MRT_2$ from
38 to 94 days, $MRT_3$ from 79 to 100 days), sites C&D ($MRT_1$ from 8 to 96 days, $MRT_2$ from 32 to 124 days, $MRT_3$ from 25
to 128 days), and site E ($MRT_1$ from 13 to 32 days, $MRT_2$ from 72 to 88 days, $MRT_3$ from 51 to 61 days) show more pronounced
differences in water source utilization. Bulk water and rock water have longer retention times, while mobile water has the
shortest retention time, averaging only 44% and 48% of the retention times of bulk water and rock water, respectively. This
suggests that at these sites, vegetation relies more on water sources with longer retention times, while the proportion of mobile
soil water, which replenishes quickly, is relatively low, possibly serving primarily to supplement short-term high transpiration
demands.

Additionally, the xylem water of trees at site $ts_6$ shows retention times for bulk water and rock water up to 124-128 days,
and mobile water up to 96 days, indicating $ts_6$'s capability to utilize long-standing water sources combined with short-term
available water to support its high transpiration demand (average sap flow rate of 1.3 cm·h$^{-1}$). This strategy maintains its high



water consumption.

Evergreen trees at sites $cc_{1-3}$ also show long retention times for bulk water and rock water (73-76 days) compared to deciduous trees, suggesting that evergreens rely more on a stable deep water supply to maintain physiological activities throughout the year.

Moreover, trees at site E primarily rely on bulk soil water, as indicated by its longer retention time (72-88 days) compared
to mobile water (13-32 days) and rock water (51-61 days), suggesting that the site's soil has strong water-holding capacity (illustrated in Figure A1c, higher SMC), providing a relatively stable water supply to the trees.

**3.3 Partitioning of plant water sources**

Based on the MixSIAR model outputs, Figs. 6a-f illustrate the proportional use of four water sources: bulk soil water, mobile soil water, rock fracture water and infilled rock fracture water by trees at different sites across various months.

During the cold and dry winter months (December-January), monthly precipitation was less than 25 mm, and tree sap flow density declined to its annual minimum, indicating limited water use by tree. As shown in Fig. 6a, however, trees still exhibited relatively high uptake of mobile soil water (30-54%, mean 39%). Previous studies have shown that even small precipitation events can temporarily elevate soil moisture content (SMC > 0.2), thereby stimulating transpiration in trees (Liu et al., 2025). This suggests that minor rainfall pulses during the dry season may be sufficient to recharge mobile water in near-
surface soils, which, despite their transient nature, can still meet short-term water demands of shallow-rooted plants or maintain baseline physiological functions in trees. In addition, at sites C&D, which have abundant rock water sources (including rock fracture water and infilled rock fracture water), trees derived 40-44% (mean 42%) of their water from these sources during the winter months.

As temperatures began to rise in February and March (early spring), tree water use also increased. During this period,
trees at sites C&D further expanded their reliance on rock water, with contributions reaching 47-88% (mean 69%) in Fig. 6b. In contrast, trees at other sites with limited rock water availability showed increased uptake of mobile soil water (43-72%, mean 57%), reflecting a greater dependence on short-term rainfall inputs. February to March represents a critical period for budburst and root activity in many tree species. These contrasting patterns suggest that in areas with well-developed rock fractures and soil-filled fractures, trees can draw upon subsurface water reserves to meet early-season water demands before
substantial rainfall resumes. Conversely, in rock water-limited environments, vegetation relies more heavily on episodic rainfall recharge of shallow mobile water to support transpiration recovery and physiological reactivation following the winter dormant period.

During the subsequent early growing season (April-May; Fig. 6c), substantial variation in water source use is observed among sites. Trees at sites A and E rely heavily on soil water (bulk and mobile), accounting for 63-91% (mean 77%) of uptake.
In contrast, trees at sites C&D show the highest dependence on infilled rock fracture water (15-74%, mean 37%). The largest tree ($ts_6$, DBH is 36 cm in Table 4) derived up to 85% of its water from rock sources, including 74% from infilled rock fracture water. This is consistent with its root zone containing the highest fracture volume among all sampled trees (2.19 m$^3$ in Table 4) of which 87-96% volume is contributed by soil-filled fractures. Table 3 shows that soil-filled fractures (≥20 mm) at C&D





have MRTs as long as 169-303 days (mean 212 days), enabling the storage of summer rainfall for gradual release into the following spring, thereby mitigating early-season soil water deficits. Additionally, Table 4 indicates that $ts_6$ uses fracture water with a residence time of up to 128 days, confirming its ability to extract long-residence water for growth and transpiration. As a large tree with an extensive and deep root system, $ts_6$ efficiently exploits deep fracture water. In contrast, smaller individuals at site E ($bp_5$, $kp_{1-2}$) receive lower contributions from rock water (9-12%), relying more on shallow soil water (< 50 cm depth).

These findings suggest that long-residence rock fracture water acts as a 'transitional reservoir', providing critical water to support tree budding and transpiration at the onset of the growing season.

During the peak rainy season (June-July; Fig. 6d), the contribution of mobile soil water increases markedly (20-68%, mean 41%), especially at sites C&D (33-68%, mean 47%) and A (49-59%, mean 53%). This shift indicates that rainfall-driven replenishment of mobile soil water becomes the primary water source for vegetation. At the same time, the average contribution of infilled rock fracture water at C&D declines from 37% (April-May) to 15%, suggesting that in the rainy season, fracture

water plays a more regulatory role, storing excess water for later use, rather than serving as the dominant source.

In August-September (Fig. 6e), the reliance on bulk water at sites C&D increases significantly from 14-27% (mean 20%) in June-July to 35-58% (mean 49%), while the use of mobile soil water drops to 10-20% (mean 16%). Combined with soil moisture data from Fig. A1c, which shows low SMC (< 0.2) in the 20-50 cm layer at C&D during this period, these trends suggest a decline in shallow soil water availability, prompting greater dependence on slowly released bulk water. In contrast,

trees at site A ($bp_{1-3}$) increase their use of mobile water to 76-82% (mean 79%), supported by relatively high soil moisture (> 0.2) at 20-50 cm depth.

During the late growing season (October-November; Fig. 6f), which marks the transition from wet to dry conditions, the proportion of bulk water used by trees generally increases (14-80%, mean 55%). As rainfall decreases significantly after October (Fig. 2) and soil moisture declines (Fig. A1c), the availability of recent precipitation diminishes, leading trees to rely

more on long-residence bulk water to meet minimal physiological demands.

Notably, trees $ts_1$ and $ts_2$ at site B continued to rely heavily on infilled rock fracture water during this period, with contributions of 39% and 58%, respectively. Similarly, trees at site A ($bp_{1-3}$) show a substantial increase in rock fracture water use: from 9-12% (mean 10%) in August-September to 43-44% in October to November. This change is consistent with the rapid depletion of soil water after the rainy season (Fig. A1c), whereas fracture water remains available due to higher fracture

moisture content (Fig. A2c). Rock water thus provides an essential supplementary source during the transition from wet to dry season, sustaining tree water use after soil water exhaustion and preventing severe water stress.

In addition, site E exhibited the least developed rock fractures (aperture < 2.6 mm; Table B3). Throughout most of the year, the trees reliance on rock water remained below 30%. However, due to the low rock permeability at this site ($K_h$ < 47 cm·h$^{-1}$; Table B3), water loss through deep percolation was limited. As a result, the average soil moisture content was relatively

high (0.204-0.298; Table B1), and trees primarily relied on soil water for transpiration.





Figure 6. Partitioning of tree water sources by month.





## 4. Discussion

### 4.1 Rock fracture properties drive isotopic variability and ecohydrological responses of vegetation

This study provides a mechanistic assessment of the role of rock fractures on plant water uptake of rock moisture. Our
work shows how fracture water can serve as a deep and temporally buffered reservoir, supporting plant water needs during
transitional periods. During both the pre-growing season and early growing season (February-May), rock water plays a critical
role in reactivating deciduous tree growth. During the peak rainy season (June-July), rainfall-driven mobile soil water
dominates plant uptake, rapidly recharging the root zone to support high transpiration. As precipitation declines in late summer
(August onward), trees increasingly rely on bulk soil water and rock water to meet their physiological demands through the
dry season.

This flexible adaptation enables vegetation to cope with seasonally dynamic and structurally complex hydrological
conditions. These results in some ways support the notion of ecohydrological separation (Brooks et al., 2010; Mcdonnell, 2014;
Evaristo et al., 2015), which posits that different subsurface water pools are functionally disconnected and differentially
accessed by plants.

This separation is primarily due to the heterogeneity of soil porous media, insufficient exchange between preferential
flow and matrix flow, and the non-uniform distribution and water use strategies of plant roots. In addition, wet-dry alternations,
rainfall intensity, and soil evaporation fractionation effects also influence this degree of separation (Sprenger et al., 2016;
Finkenbiner et al., 2022). Conditions such as low soil water content, preferential flow paths, and coarse-textured, low-
saturation soils with poor hydraulic conductivity all act to reinforce ecohydrological separation (Liu et al., 2020; Finkenbiner
et al., 2021; Finkenbiner et al., 2022). The structural complexity of the karst subsurface, including variations in fracture
aperture, depth, and degree of soil infill, contributes to the limited mixing among these water pools and reinforces their
functional separation.

Our study reveals significant isotopic variability in fracture water across different depths and fracture apertures (Figs.
A4a and A4b). In the 0-200cm depth range, the Lc-excess values of fracture water are close to 0‰, with many values even
falling below 0‰, indicating that this range of fracture water receives more recent rainfall replenishment and experiences
weaker evaporation effects. However, deep fracture water at depths greater than 200 cm typically has Lc-excess values above
0‰, suggesting that deep fracture water has undergone a longer storage and vapor exchange process, with water sources being
long-term precipitation retention or underground replenishment. Additionally, the aperture of the fractures also significantly
affects the isotopic characteristics of the water. Fracture water with apertures between 0.5 and 2 mm exhibits a wide range of
$\delta D$ and $\delta^{18}O$, exceeding the isotopic variation range of xylem water in vegetation. This indicates that the water in these fractures
has undergone more complex replenishment and migration processes influenced by soil water, precipitation infiltration, and
evaporation fractionation. Meanwhile, fracture water with apertures between 10 and 38 mm has isotopic values within the
range of xylem water, and its Lc-excess is less than 0‰—closer to xylem water, suggesting that such fracture water serves as
a stable water source for vegetation.

Fracture water at 600 cm depth shows more concentrated $\delta D$ and $\delta^{18}O$ values, with a much narrower isotopic range than





xylem water. Therefore, if this deep fracture water is used to represent root zone fracture water for vegetation water source partitioning, it may fail to explain the entire water utilization strategy of vegetation and lead to errors. Many studies that use deep spring water (Rong et al., 2011; Nie et al., 2012; Deng et al., 2015; Carrière et al., 2020; Ding et al., 2021; Wu et al., 2021; Zeng et al., 2021; Cai et al., 2023; Fan et al., 2023; Wu et al., 2024b; Cai et al., 2025), or mixed borehole water (Deng et al., 2020) as epikarst zone water for vegetation water source partitioning are not rigorous. Our findings suggest that research on vegetation water use in karst regions should consider the impact of different fracture depths and apertures, as the heterogeneity of fracture spaces can lead to significant variations in isotopic composition.

Our findings suggest that accurate analysis of plant water use and ecohydrological separation in karst systems must explicitly incorporate rock water into the water source framework. This is to account for seasonal water source shifts under varying hydroclimatic conditions and to fully understand the functioning of the coupled soil-plant-rock fracture continuum.

## 4.2 On the surprising similarity between mobile soil water and bulk soil water

Our isotopic correlations between bulk soil water and mobile soil water. Fig. A5 shows that during dry seasons (February-May and October-January), the correlation between the isotopic values of bulk soil water and mobile water is weak ($R^2$ = 0.002-0.34). This indicates poor hydraulic coupling between the two components. Notably, the isotopic values of bulk water (obtained using the CVD method) are more depleted than those of mobile water (using suction lysimeters). This is consistent with the initial findings of Brooks et al. (2010) and follow-up work by Sprenger et al. (2019), Zhao and Wang (2021) and Xu et al. (2025). Brooks et al. (2010) conducted their study in a Mediterranean climate within the western Cascade Mountains of Oregon, USA. In the H. J. Andrews Experimental Forest, they collected samples during the dry season (June, August, September 2004-2005) and the wet-up period in autumn 2006 (October-December). Across these campaigns, Brooks et al. (2010) found that the weak isotopic correlation between bulk and mobile water suggests limited mixing, likely resulting from their residence in different pore domains. This implies that during these periods, bulk and mobile water are influenced by distinct recharge sources and subject to different degrees of evaporation-driven isotopic fractionation, thereby enhancing ecohydrological separation.

However, in our study, we observed that during the high-rainfall period in June- September, this separation effect becomes less pronounced. The correlation between bulk and mobile soil water isotopic values improves markedly ($R^2$ = 0.60-0.61; Fig. A5), indicating that recent precipitation inputs increasingly dominate the mobile water pool and that some degree of hydraulic exchange occurs between the two compartments. This seasonal shift suggests a dynamic and partial coupling between soil water pools under high moisture conditions, which may temporarily weaken ecohydrological separation (Geris et al., 2015). Snelgrove et al. (2021) report that in a catchment in northern Britain, high soil water storage and limited precipitation inputs in humid settings can facilitate mixing between soil water and mobile water, reducing the likelihood of pronounced separation. Similarly, in ecosystems with high rainfall and elevated soil moisture, such as tropical rainforests, the isotopic differences between soil water and plant xylem water are often small or even negligible, indicating little to no separation (Liu et al., 2020). Ecohydrological separation is more likely during dry periods and tends to weaken or disappear during the wet season (Hervé-



Fernández et al., 2016; Luo et al., 2019).

To further investigate the hydraulic linkage of different water components at varying depths, we analyze isotopic relationships stratified by depth. The results show that in the upper 0-50 cm layer, the correlation between soil bulk and mobile water is relatively strong ($R^2$ = 0.68-0.78), whereas in the 50-200 cm layer, the correlation weakens significantly ($R^2$ = 0.32-0.46; Fig. 7). Vargas et al. (2017) demonstrated that under wet conditions, 75-95% of mobile water can exchange isotopically with surrounding bound water in the upper soil layers (< 30 cm).

Fig. A6 further reveals that with increasing depth, mobile soil water becomes increasingly enriched in $\delta D$ and $\delta^{18}O$, while its isotopic range remains relatively stable, and Lc-excess values remain near 0‰. In contrast, bulk soil water shows a gradual depletion in $\delta D$ and $\delta^{18}O$ with depth, along with significantly lower Lc-excess values, aligning more closely with xylem water.

     These findings suggest that in surface soils, precipitation infiltrates rapidly and alters the isotopic composition of both mobile and bulk soil water, while also enhancing hydraulic connectivity between different pore domains. This is particularly

evident during periods of intense rainfall, when high soil moisture content reduces matric potential gradients and facilitates lateral and vertical water exchange across pore domains. Under such conditions, even tightly bound water may undergo partial displacement or mixing due to sustained infiltration and elevated hydraulic gradients.

     With increasing depth, however, the hydraulic connection between bulk and mobile water progressively weakens. Deep bulk water increasingly behaves as a relatively isolated reservoir, with limited exchange, while mobile water remains

hydrologically dynamic and responsive to recent precipitation inputs through preferential flow pathways. This vertical stratification in hydraulic behavior results in the clear manifestation of ecohydrological separation within the soil profile under most seasonal conditions.

     Nevertheless, during the peak rainy season, the high volume and intensity of precipitation enhance water fluxes and recharge rates throughout the profile. This leads to greater pore connectivity and transient mixing between bulk and mobile

water pools, thereby weakening the separation effect. Thus, our findings demonstrate that ecohydrological separation is not static but modulated by seasonal hydrological regimes, with stronger separation under drier conditions and partial coupling during wetter periods when hydrodynamic forces override structural constraints.





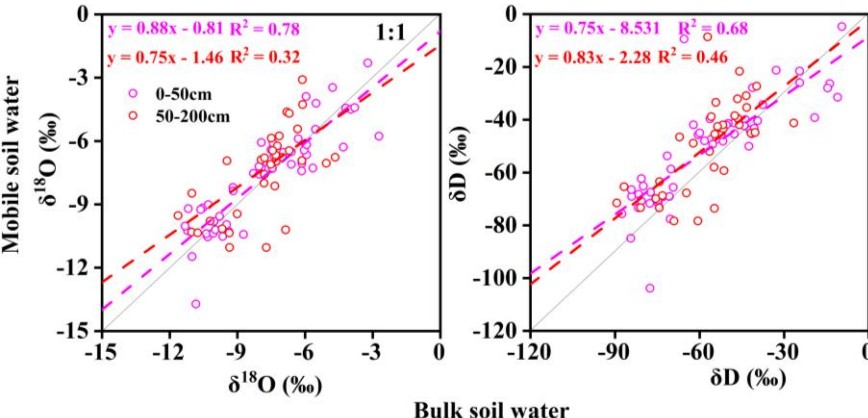

**Figure 7. Isotopic comparisons (δD and δ¹⁸O) between bulk soil water and mobile water across soil depths.**

### 4.3 A perceptual model for the role of rock fractures in plant water uptake in karst

Our findings lead to a perceptual model that illustrates how plants dynamically access different water sources in karst ecosystems, shaped by the structural and hydrodynamic properties of the soil-rock continuum. The conceptual framework

presented in Figure 8 depicts a vertically stratified water source system composed of four primary compartments: mobile soil water, bulk soil water, rock fracture water, and infilled rock fracture water.

Overall, the preferential water source hierarchy in this karst system follows the order: mobile soil water > bulk soil water > rock water. In our study, mobile water is characterized by short residence times and high turnover. Mobile soil water is the primary support for transpiration during wet periods but is less accessible during dry spells. In contrast, bulk soil water, with

its longer MRT, is mainly used by vegetation during the late growing season. Rock water, especially that stored in soil-filled fractures, with its longer MRT and higher retention capacities, acts as a more stable source, sustaining plant function when soil water is depleted, thereby substantially enhancing tree survival and transpiration stability during drought periods and facilitating the onset of the growing season (Schwinning, 2020; Hahm et al., 2022; Leite et al., 2025).





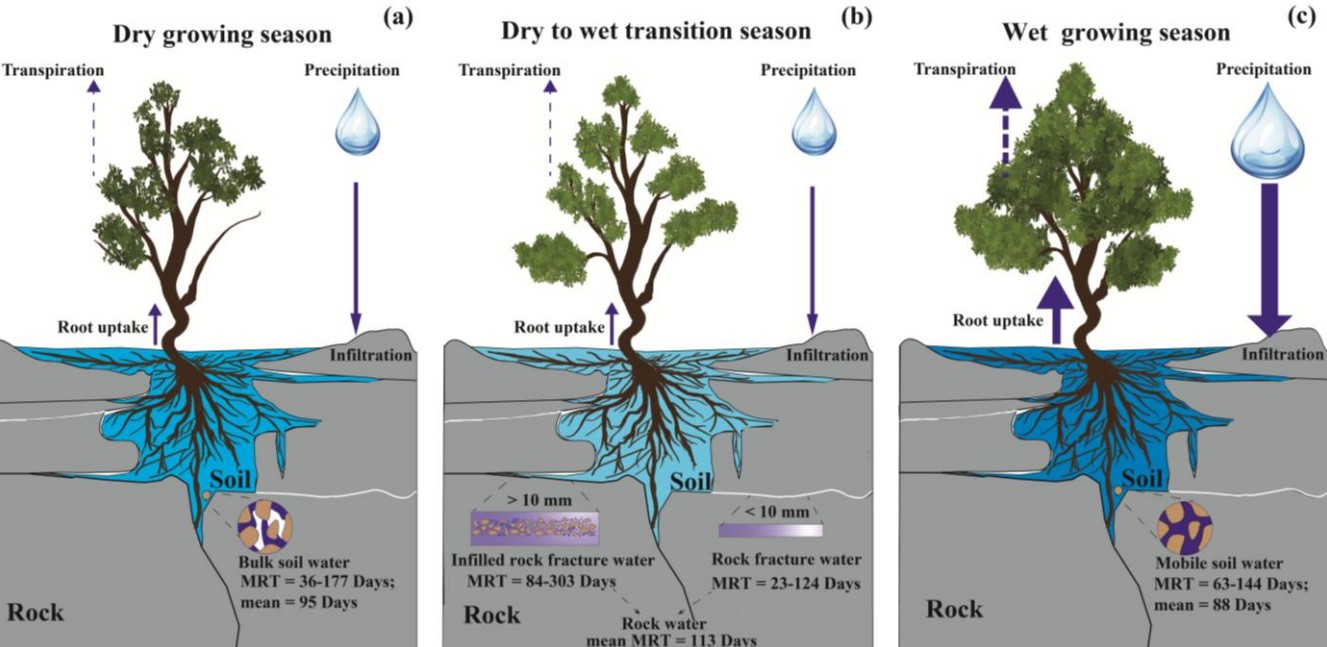

**Figure 8. Perceptual model of the plant water uptake in karst.**

Specifically, precipitation infiltrates the thin soil layer and preferentially recharges mobile soil water stored in connected pores. This water is readily available for root uptake but has a short residence time and low retention capacity, particularly during the wet season. As water percolates downward, part of it is retained in micropores as bulk soil water, which is replenished more slowly and supports vegetation during moderately dry periods. At greater depths, plant roots interact with the weathered bedrock. Rock fractures with smaller apertures and minimal soil infill function as transient water storage compartments with intermediate MRT, whereas larger fractures that are substantially filled with soil serve as long-term reservoirs, characterized by significantly prolonged water retention (MRT up to 303 days in our study). These fractures are crucial for maintaining transpiration during dry or transitional periods (e.g., early spring and late fall), especially for large, deep-rooted trees. Our isotope and MRT analyses show that vegetation flexibly shifts water use based on seasonal availability, progressively relying on infilled rock fracture water and bulk soil water as mobile soil water becomes scarce.

In Southwest China, long-term anthropogenic disturbances and natural processes have driven extensive karst desertification, resulting in severe soil degradation, vegetation loss, and ecosystem instability (Wang et al., 2019b). Despite a general "greening" trend since 2000 (Tong et al., 2018), projections indicate that drought frequency in Southwest China is likely to increase further between 2021 and 2035 (Climate Change Assessment Report for Southwest China, 2021). A combination of reduced precipitation, rising temperatures, more frequent and intense droughts, soil degradation, and insufficient water resource management may amplify the vulnerability of current forest restoration efforts (Lin et al., 2015; Peng et al., 2020; Zhou et al., 2020; Zhao et al., 2024), affecting plant water uptake and long-term ecosystem stability. With global warming intensifying the hydrological cycle, shifts between wet and dry periods are becoming more pronounced (He



et al., 2024; Chen et al., 2025). It is projected that by 2040-2100, nearly 60% of the global land surface will experience accelerated dry-wet transitions (Chen and Wang, 2022). The frequency of both atmospheric drought and surface soil drought is rising (Fabiani et al., 2024; Xu et al., 2024), which may lead to fundamental changes in plant water-use strategies (Salomón et al., 2022; Wei et al., 2023). Under such conditions, rock fracture water may serve as a crucial buffer, sustaining vegetation
during drought events (Korboulewsky et al., 2020; Schwinning, 2020; Luo et al., 2024b; Wu et al., 2024a). However, current research has not adequately elucidated the regulatory mechanisms by which soil-rock structures control water transport and retention times, nor their influence on vegetation water uptake. In particular, there is a lack of quantitative, mechanistic studies focusing on factors such as fracture aperture, soil-filled fractures, soil physical properties, and the interactions between vegetation and the soil-rock structure.

More broadly, our framework highlights the underappreciated role of rock fracture water in sustaining vegetation function under increasing climate variability. Future ecohydrological modeling and karst forest restoration should explicitly incorporate the structure and hydrodynamics of the soil-rock continuum to improve resilience and adaptive capacity in these fragile landscapes.

## 5. Conclusion

Our study represents one of the first mechanistic investigations into the role of rock fractures in facilitating rock moisture uptake by plants in karst environments. By integrating stable isotope analysis with MRT modeling, we delineated four functionally distinct water compartments within a structurally heterogeneous soil-rock continuum: mobile soil water, bulk soil water, rock fracture water, and infilled rock fracture water. Our results demonstrate that plant water uptake strategies are seasonally modulated by the availability and hydrodynamic behavior of these compartments. Notably, vegetation exhibits a
clear preference for water sources with shorter MRTs, especially under wetter conditions. During the wet season, rainfall-driven mobile soil water predominates in supporting plant transpiration. In contrast, during dry or transitional periods, trees increasingly relies on bulk soil water and rock fracture water, particularly from infilled fractures characterized by longer MRTs (84-303 days). The limited mixing and distinct MRTs observed among these water pools provide strong support for the ecohydrological separation hypothesis. However, under high moisture conditions, increased hydraulic connectivity appears to
reduce this separation. The perceptual model developed from this work highlights the critical buffering role of rock fracture water in sustaining vegetation under variable hydroclimatic regimes. These findings underscore the importance of incorporating bedrock heterogeneity, water source dynamics, and seasonal shifts in plant water use into ecohydrological models of karst systems. Given the anticipated increases in drought frequency and climatic variability, future forest management and restoration strategies in karst landscapes should explicitly consider the integrated behavior of the soil-rock-
plant continuum to enhance ecosystem resilience.



**Appendix A**



**Figure A1.** Dynamic variations in stable isotopes of soil water at different depths at sites (A-E): (a) δ¹⁸O and (b) δD, and moisture
content: (c) and (d). Mobile soil water (MSW), bulk soil water (BSW), soil-rock interface water (SRW), soil moisture content
(SMC), soil-rock interface moisture content (SRMC).





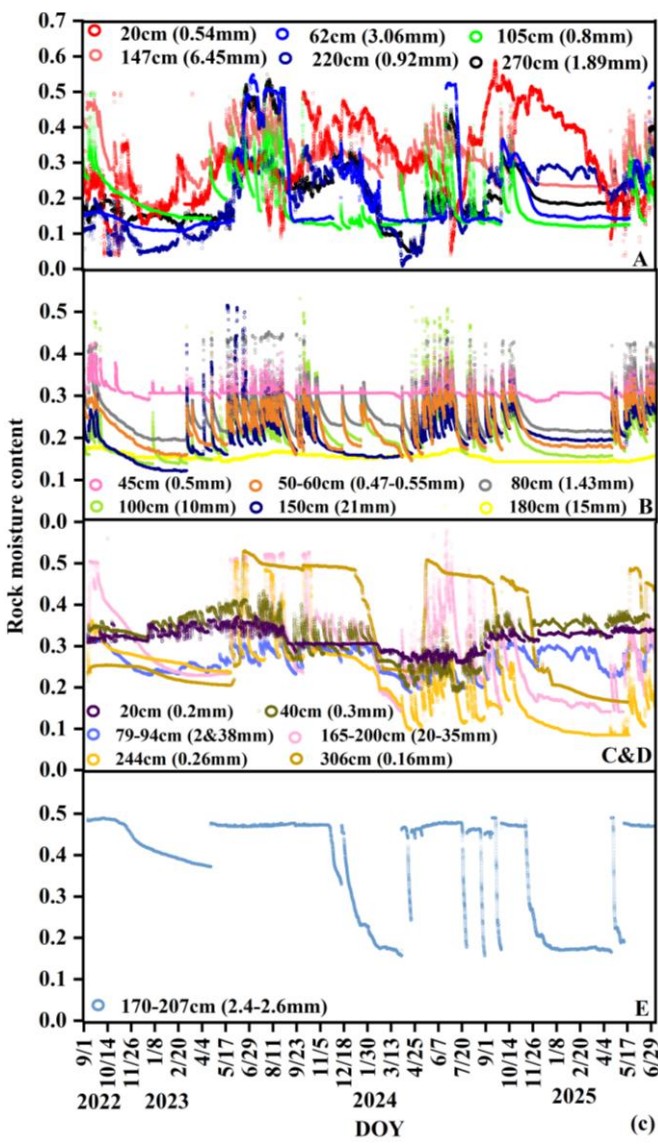

**Figure A2. Dynamic variations in stable isotopes of rock fracture water at different depths / apertures at sites (A-E): (a) δ¹⁸O and (b) δD, and moisture content: (c).**





**Figure A3. Dynamic variations in stable isotopes of xylem water (δD and δ¹⁸O) in different trees at sites (A-E).**







**Figure A4. Average values and ranges of δD, δ¹⁸O and Lc-excess in rock waters with different apertures.**



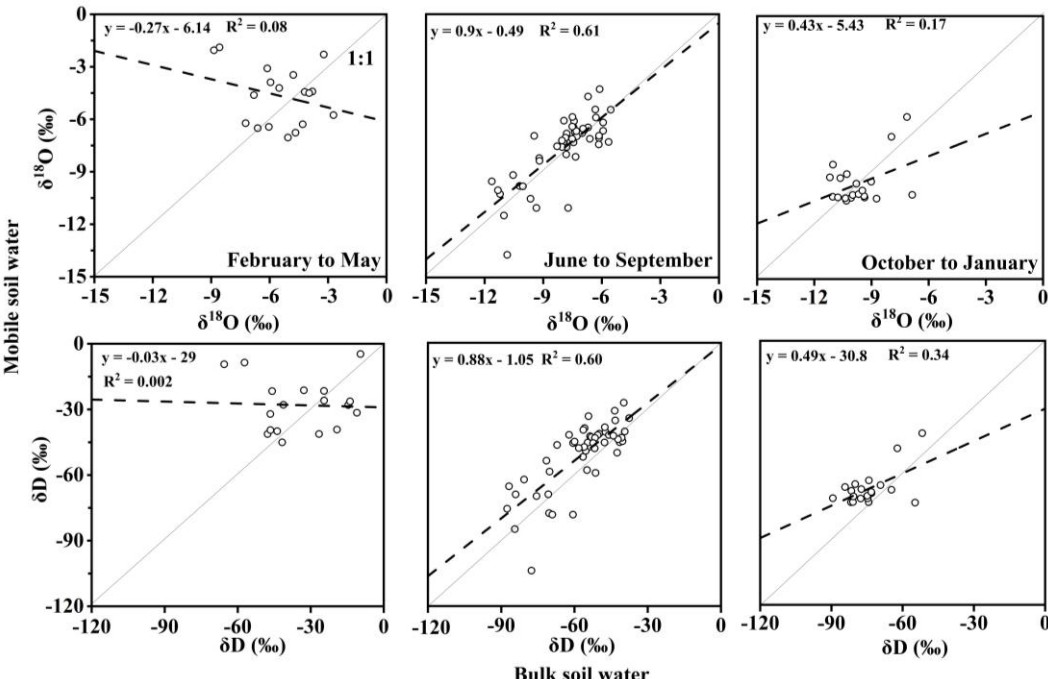

**Figure A5. Isotopic comparisons (δD or δ¹⁸O) between bulk soil water and mobile soil water.**

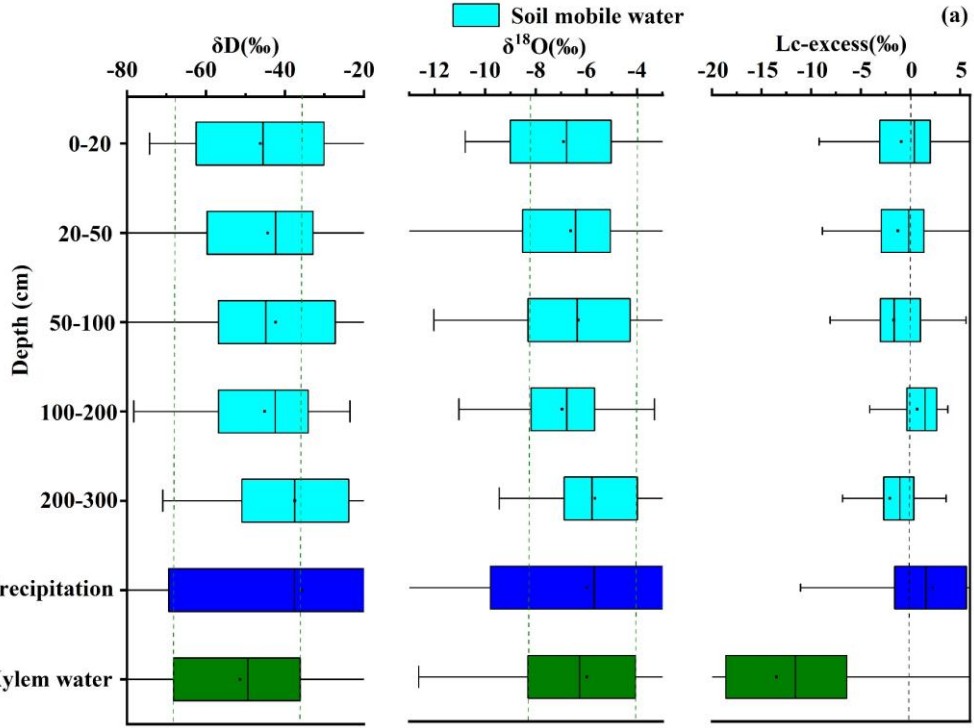



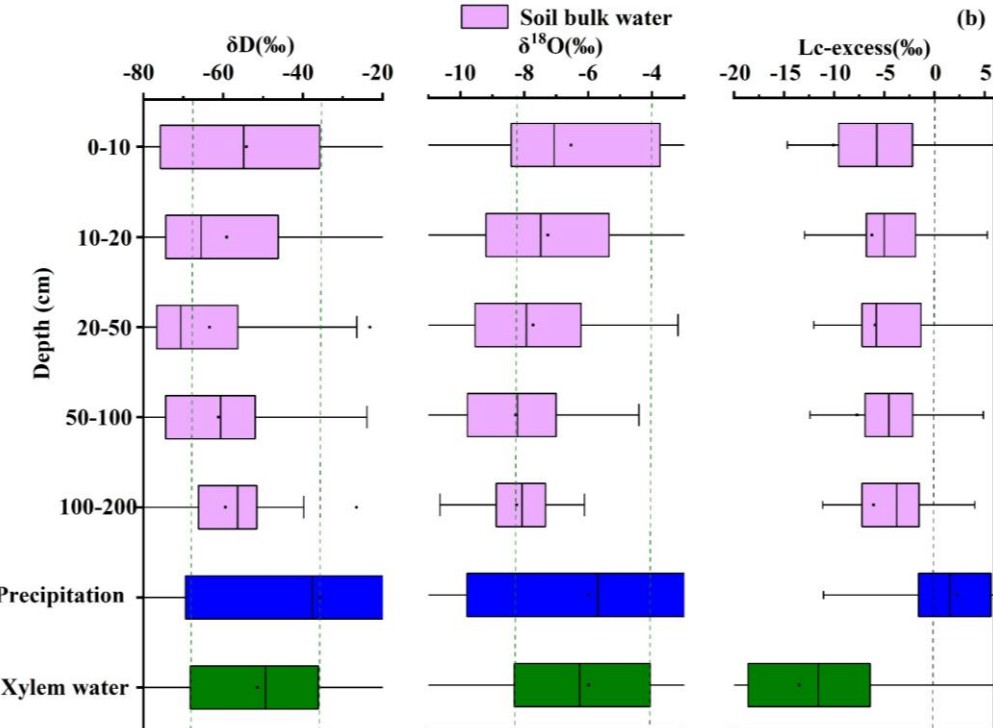

**Figure A6. Average values and ranges of δD, δ¹⁸O and Lc-excess in different soil waters.**




**Appendix B**

To distinguish it from mean residence time (MRT), we applied the mean transit time (MTT) phase shift method from
Allen et al. (2018) to quantify the time delay in water transport between compartments. The phase shift primarily captures the
time lag between input signals (e.g., precipitation isotopic composition) and output signals (e.g., plant or groundwater isotopic
composition). MTT represents the time lag or transmission time between different compartments in the hydrological cycle
(e.g., precipitation to soil, soil to rock, rock to plant). The MTT (in day) calculation was made by:

$$MTT = c^{-1}(\phi_w - \phi_p) \tag{B1}$$

Where $c = 2\pi \cdot 365^{-1}$ is the angular frequency constant, which represents the yearly cycle (assuming a one-year period for
seasonal variations); $\phi_p$ is the phases of precipitation sine curves, and $\phi_w$ is water sine curve (mobile soil water, bulk soil water,
rock fracture water, infilled rock fracture water, soil-rock interface water and xylem water).

Table B1 shows MTT was positively correlated with MRT (MTT = 19.55ln(MRT) - 66, $R^2$ = 0.77), with a range of 6-82
days and an average of 28 days. Site B also exhibited significantly higher MTT values (10-82 days, mean 39 days) compared
to other sites (6-38 days, mean 21 days).

**Table B1. Statistics of MRT and MTT (day) for mobile and bulk water in different soil layers.**

| Depth (cm) | Site | $K_h$ (cm·h⁻¹) | Average moisture content | Mobile soil water MRT | Mobile soil water MTT | Bulk soil water MRT | Bulk soil water MTT |
|---|---|---|---|---|---|---|---|
| 5 | | 3.83 | 0.153 | - | - | 36 | 6 |
| 20 | | 3.33 | 0.223 | - | - | 65 | 16 |
| 50 | A | 2.33 | 0.264 | 72 | 15 | 139 | 26 |
| 70 | | 2.18 | - | - | - | 155 | 28 |
| 100 | | 1.96 | 0.133 | 74 | 31 | 138 | 32 |
| 5 | | 0.58 | 0.212 | - | - | 49 | 10 |
| 20 | | 0.44 | 0.245 | 63 | 23 | 177 | 31 |
| 50 | B | 0.16 | 0.241 | 459 | 34 | 471 | 73 |
| 70 | | 0.17 | - | - | - | 638 | 82 |
| 100 | | 0.18 | 0.125 | 244 | 43 | 685 | 47 |
| 180 | | 0.1 | 0.182 | 77 | 24 | 708 | 63 |
| 5 | | 4.96 | 0.191 | - | - | 40 | 6 |
| 20 | | 1.04 | 0.261 | 85 | 25 | 60 | 12 |
| 30 | C&D | 1.04 | - | - | - | 66 | 9 |
| 40 | | 1.04 | 0.209 | 85 | 33 | 63 | 12 |
| 50 | | 1.04 | 0.209 | 63 | 20 | 95 | 17 |
| 5 | | 1.42 | 0.309 | - | - | 47 | 8 |
| 30 | | 0.75 | 0.227 | 122 | 38 | 108 | 19 |
| 50 | E | 0.75 | 0.298 | - | - | 131 | 27 |
| 70 | | 0.83 | - | - | - | 118 | 24 |
| 100 | | 0.96 | 0.204 | 144 | 32 | 121 | 17 |





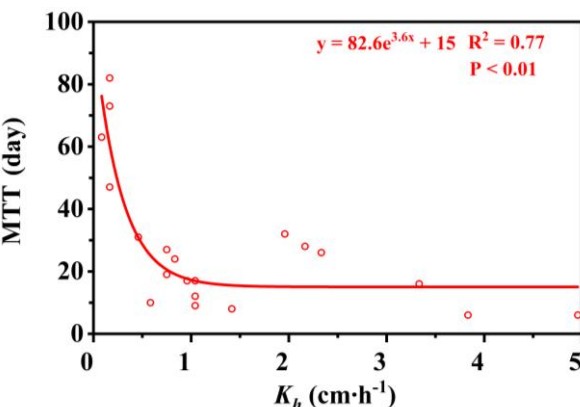

**Figure B1. Relationship between $K_h$ and MTT of bulk soil water.**

**Table B2. Statistics of MRT and MTT (day) for soil-rock interface water in different soil layers**

| Depth (cm) | Site | Average moisture content | MRT | MTT |
|---|---|---|---|---|
| 50 | | 0.254 | 119 | 39 |
| 100 | B | 0.166 | 21 | 25 |
| 180 | | 0.154 | 214 | 41 |

**Table B3. Statistics of MRT and MTT (day) for fractures of different apertures and depths.**

| Aperture (mm) | Depth (cm) | Site | Porosity ($\varepsilon_p$) | $K_h$ (cm·h$^{-1}$) | Average moisture content | MRT | MTT |
|---|---|---|---|---|---|---|---|
| 0.16 | 306 | C&D | 1 | 0.09 | 0.354 | 124 | 31 |
| 0.26 | 244 | C&D | 1 | 1.13 | 0.256 | 118 | 34 |
| 0.47 | 60 | B | 1 | 3.04 | 0.217 | 57 | 24 |
| 0.50 | 45 | B | 1 | 3.42 | 0.310 | 23 | -2 |
| 0.55 | 50 | B | 1 | 3.71 | 0.217 | 91 | 25 |
| 1.43 | 80 | B | 0.98 | 17 | 0.260 | 76 | 22 |
| 1.8 | 248 | E | 0.75 | 13 | - | 73 | 20 |
| 1.89 | 270 | A | 0.98 | 673 | 0.217 | 108 | 30 |
| 2 | 79 | C&D | 0.74 | 76 | 0.256 | 35 | 2 |
| 2.4 | 170 | E | 0.6 | 34 | 0.413 | 103 | 29 |
| 2.6 | 207 | E | 0.54 | 47 | 0.413 | 97 | 33 |
| 3.06 | 62 | A | 0.80 | 40 | 0.203 | 60 | 22 |
| 5 | 600 | A | 0.66 | 3496 | - | 182 | 29 |
| 6.45 | 147 | A | 0.55 | 6208 | 0.317 | 100 | 24 |
| 10 | 100 | B | 0.41 | 6 | 0.206 | 136 | 27 |
| 15 | 180 | B | 0.45 | 33 | 0.158 | 84 | 21 |
| 20 | 200 | C&D | 0.47 | 315 | 0.311 | 176 | 25 |
| 21 | 150 | B | 0.74 | 1042 | 0.188 | 136 | 27 |
| 30 | 165 | C&D | 0.49 | 87 | 0.311 | 303 | 20 |
| 35 | 176 | C&D | 0.74 | 2232 | 0.311 | 169 | 38 |
| 38 | 94 | C&D | 0.70 | 2746 | 0.256 | 200 | 24 |



**Table B4. Statistics of MRT and MTT (day) for different trees at typical sites in the study area.**

| Site | Number | Diameters (cm) | Canopy area (m²) | Soil depth (cm) | Soil volume (m³) | Fracture volume (m³) | Average sap flow density (cm·h⁻¹) | From mobile soil water to tree | | From bulk soil water to tree | | From rock water to tree | |
|---|---|---|---|---|---|---|---|---|---|---|---|---|---|
| | | | | | | | | $MRT_1$ | $MTT_1$ | $MRT_2$ | $MTT_2$ | $MRT_3$ | $MTT_3$ |
| A | $bp_1$ | 29.6 | 62.83 | 200 | 1.68 | 1.30 | 1.59 | 53 | -1 | 58 | 5 | 47 | -4 |
| | $bp_2$ | 12.8 | 25.04 | 200 | 0.67 | 0.52 | 0.36 | 59 | 5 | 64 | 10 | 53 | 1 |
| | $bp_3$ | 22 | 47.12 | 200 | 1.26 | 0.98 | 1.20 | 39 | -8 | 44 | -3 | 31 | -12 |
| B | $ts_1$ | 5.4 | 1.30 | 43 | 1.12 | 0.04 | 1.03 | 14 | 0 | 38 | -13 | 79 | 3 |
| | $ts_2$ | 5.8 | 1.51 | 22 | 0.42 | 0.05 | 1.09 | 40 | -7 | 94 | -3 | 101 | -3 |
| | $ts_3$ | 10 | 10.40 | 24 | 0.22 | 1.01 | 0.22 | 18 | -9 | 60 | 8 | 54 | 7 |
| | $ts_4$ | 9.5 | 13.70 | 60 | 0.32 | 1.33 | 1.11 | 22 | -10 | 48 | 6 | 51 | 7 |
| | $ts_5$ | 10.77 | 14.00 | 16 | 0.12 | 0.61 | 0.41 | 8 | -9 | 32 | 7 | 25 | 7 |
| | $ts_6$ | 36 | 50.46 | 60 | 6.88 | 2.19 | 1.30 | 96 | 2 | 124 | 18 | 128 | 19 |
| C&D | $cc_1$ | 13.4 | 17.01 | 24 | 0.22 | 1.65 | 1.25 | 42 | -4 | 79 | 13 | 73 | 12 |
| | $cc_2$ | 14.39 | 15.47 | 30 | 0.68 | 1.50 | 0.80 | 43 | -1 | 80 | 15 | 74 | 14 |
| | $cc_3$ | 11.73 | 21.90 | 52 | 0.71 | 0.95 | 1.02 | 45 | 2 | 68 | 19 | 71 | 19 |
| | $yd_1$ | 9.63 | 10.07 | 53 | 0.37 | 0.44 | 0.93 | 37 | -15 | 60 | 1 | 63 | 2 |
| | $yd_2$ | 12.75 | 7.52 | 52 | 1.42 | 0.33 | 1.04 | 29 | -12 | 54 | 4 | 57 | 5 |
| E | $bp_4$ | 16 | 28.04 | 100 | 2.40 | 0.38 | 0.50 | 13 | -21 | 72 | -1 | 61 | -13 |
| | $bp_5$ | 13.3 | 30.68 | 15 | 0.15 | 0.41 | 1.72 | 26 | -21 | 83 | 5 | 52 | -11 |
| | $kp_1$ | 8 | 4.24 | 20 | 0.19 | 0.06 | 0.88 | 25 | -33 | 83 | -6 | 51 | -23 |
| | $kp_2$ | 6.8 | 2.83 | 20 | 0.18 | 0.04 | 1.16 | 32 | -22 | 88 | 5 | 56 | -12 |

*Data availability.*

To validate the results of this study and ensure reproducibility, the datasets are accessible on Zenodo (https://zenodo.org/records/14827769, https://zenodo.org/records/16810833) and are publicly available for download (Liu, 2025a, b).

*Author contributions.*

XL, XC, and JJM conceived the study. XL, WL, and ZZ curated the data. XL and JJM performed the formal analysis. XL and WL carried out the investigation. XL, XC, and JJM developed the methodology. XC, TP, and JJM provided resources. XL developed the software and performed the validation. XL and JJM prepared the original draft. XL, XC, ZZ, and JJM reviewed and edited the manuscript.

*Competing interests.*

The contact author has declared that none of the authors has any competing interests.

*Disclaimer.*

Publisher's note: Copernicus Publications remains neutral with regard to jurisdictional claims in published maps and institutional affiliations.

*Acknowledgments.*

We thank Kim Janzen for conducting all isotopic analyses, and Cody Millar and Hongxiu Wang for their assistance with sample extractions at the University of Saskatchewan. We also gratefully acknowledge Bingcheng Si, Qin Liu and Ha Fu for their valuable comments on the manuscript. The Department of Soil Science at the University of Saskatchewan is thanked for





hosting the senior author while the paper was prepared. We thank the Puding Karst Ecosystem Research Station, Chinese

Academy of Sciences, for providing field experimental support, and Dr. Jia Chen for kindly sharing the meteorological data. This work was supported by the China Scholarship Council (Grant No. 202306250097).

*Financial support.*

This research was funded by the National Natural Science Foundation of China (42030506, 42261144672).

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
