# Peer review of "The role of rock fractures on tree water use of water stored in bedrock: Mixing and residence times"

_EGUsphere, 2025_

## Referee Comment (RC1)

**Comment on egusphere-2025-3937**

This study employs a combination of stable isotope tracing, the MixSIAR Bayesian mixing model, and hydrometric observations to quantitatively analyze the spatiotemporal variations in tree water sources within karst regions, and to explore the role of rock fractures in tree utilization of bedrock-stored water. The research topic is of considerable scientific significance and shows a reasonable degree of originality, providing valuable insights into plant water-use strategies and ecohydrological coupling mechanisms under complex lithological conditions. Overall, the study presents a clear research framework and is supported by adequate data; however, improvements are needed in the description of experimental details, the clarity of figure presentation, and the logical interpretation of results. I recommend a major revision before the manuscript can be considered for publication.

**Comments**

- 1. L130-133: It is recommended to present the sampling information in a table, including details such as the number of samples and sampling frequency. In addition, the current description of different depths is rather vague.
- 2. L243-245: After heavy rainfall events, precipitation isotopes may become depleted, and in addition to the contribution of high-altitude water vapor, potential mechanisms may include enhanced convective processes leading to increased water vapor mixing, reduced evaporation of raindrops during their descent, and changes in air mass trajectories and water vapor sources. The relative contributions and magnitudes of these mechanisms still require further analysis.
- 3. L245-248: Under the influence of continental air masses, precipitation isotope values may indeed be relatively high, but "evaporation" may not be the primary cause. More critical factors likely include low precipitation amounts, short air mass transport paths, and low temperatures, which reduce the effectiveness of isotope fractionation.
- 4. L253-254: Regarding the average values of precipitation isotopes, please clarify whether a simple arithmetic mean or a weighted mean was used, and note that the same applies to soil water isotopes.

- 5. L260-263: What is the underlying mechanism for these variations? Possible factors include isotope enrichment caused by evapotranspiration, isotope depletion resulting from precipitation input, and potentially the effects of soil water storage and mixing.
- 6. L265: Here, it would be more appropriate to use the term "soil water line" rather than "evaporation line".
- 7. L268-269: The manuscript mentions "relatively stronger evaporative enrichment," but when  $\delta D$  is depleted, it typically reflects source water influence or evaporation-induced depletion rather than enrichment; the authors should rephrase this statement and clarify the mechanisms driving the variations in  $\delta^{18}O$  and  $\delta D$ .
- 8. L293-294: It is recommended to mention here that plant water uptake may also be influenced by factors such as preferential uptake, root depth distribution, and soil water availability.
- 9. L355–357: MRT is typically derived from isotope-based modeling as the "mean residence time." Can it be directly used to distinguish between "root uptake delay" and "within-tree storage"? Is the author actually referring to MTT (mean transit time) here?
- 10. L375-384: The MixSIAR model inherently involves uncertainty, and does the 30–54% mentioned here represent the uncertainty interval?
- 11. L400: MRT represents a statistical mean reflecting the average turnover rate of a water pool, and it is debatable whether it can be directly equated to "how much seasonal precipitation a water body can store and release precisely" The concept and applicability of MRT should be clarified in the manuscript to avoid potential misunderstandings.
- 12. L461: Lc-excess primarily reflects the intensity of water evaporation and the characteristics of water recharge, nor can it be directly used to indicate water storage time.

**Minor Comments**

- 1. Fig. 1 requires improvement, with more detailed information on the study area and sampling locations.
- 2. Fig. 2 has low readability, and splitting it into two panels for meteorological and

isotope data is recommended.

- 3. L253: Does the term "precipitation" in the manuscript refer exclusively to rainfall, and if not, please clarify the type of precipitation.
- 4. Fig.3 lacks a legend indicating rain water.
- 5. L264-265: The statement "average  $\delta D$  is between 59.61%" appears to be a error.
- 6. It is recommended to use colors with higher contrast in Fig. 7 for clearer display of the comparison results.
- 7. Check the legends of Fig. A1 and Fig. A2 for accuracy, and ensure the readability of the figures. Adjust the legend and text sizes if necessary.
- 8. Some of the parameters in the supplementary tables are missing units.